# Involvement of the Expression of G Protein-Coupled Receptors in Schizophrenia

**DOI:** 10.3390/ph17010085

**Published:** 2024-01-09

**Authors:** Raluka Kalinovic, Andrei Pascariu, Gabriela Vlad, Diana Nitusca, Andreea Sălcudean, Ioan Ovidiu Sirbu, Catalin Marian, Virgil Radu Enatescu

**Affiliations:** 1Doctoral School, University of Medicine and Pharmacy Victor Babes Timisoara, 300041 Timisoara, Romania; 2Eduard Pamfil Psychiatric Clinic, Timisoara County Emergency Clinical Hospital, 300425 Timisoara, Romania; andrei.pascariu@umft.ro (A.P.); gabriela.vlad@umft.ro (G.V.); enatescu.virgil@umft.ro (V.R.E.); 3Department of Biochemistry, University of Medicine and Pharmacy Victor Babes Timisoara, 300041 Timisoara, Romania; nitusca.diana@umft.ro (D.N.); ovidiu.sirbu@umft.ro (I.O.S.); cmarian@umft.ro (C.M.); 4Center for Complex Networks Science, Victor Babes University of Medicine and Pharmacy, Pta Eftimie Murgu Nr. 2, 300041 Timisoara, Romania; 5Discipline of Sociobiology, Department of Ethics and Social Sciences, George Emil Palade University of Medicine, Pharmacy, Science and Technology of Târgu Mureș, 540136 Targu Mures, Romania; andreea.salcudean@umfst.ro; 6Discipline of Psychiatry, Department of Neurosciences, University of Medicine and Pharmacy Victor Babes Timisoara, 300041 Timisoara, Romania

**Keywords:** schizophrenia, G protein-coupled receptors, psychosis, cognitive deficit, super conserved receptor expressed in the brain

## Abstract

The expression of GPCRs has been associated with schizophrenia, and their expression may induce morphological changes in brain regions responsible for schizophrenia and disease-specific behavioral changes. The articles included in this review were selected using keywords and databases of scientific research websites. The expressions of GPRs have different involvements in schizophrenia, some increase the risk while others provide protection, and they may also be potential targets for new treatments. Proper evaluation of these factors is essential to have a better therapeutic response with a lower rate of chronicity and thus improve the long-term prognosis.

## 1. Introduction

Schizophrenia is a debilitating disease that is characterized by delusions, hallucinations, abnormal behavior, and affective and cognitive impairments. An important aspect of the pathophysiology of the disease is the dysfunction of neurotransmitter systems, such as dopamine and glutamate, especially in the hippocampus [1]. 

Schizophrenia is thought to be the result of a complex interaction between genetic and environmental factors that influence early brain development and the trajectory of biological adaptation to life experiences. Genetic risk factors account for about 4/5 of the risk of developing this disease [2], and their dynamics are currently incompletely elucidated. Environmental factors can exert their influence during the prenatal (maternal neuroinfections during pregnancy), perinatal (infections and peripartum fetal hypoxia), and postnatal periods (urban environment, consumption of psychoactive substances, and others) [3]. Regarding the morphological substrate, the brains of patients with schizophrenia have certain anatomical features compared to the brains of healthy subjects, these being a reduced brain volume, especially in the temporal areas and prefrontal lobes, an increase in the size of the cerebral ventricles, and changes in the hippocampus [4,5]. Through magnetic resonance imaging studies, changes were also identified in certain regions of the limbic system [6]. At the microscopic level, a change of the normal cerebral cytoarchitecture was found, with the neurons being small, compact, and with reduced arborizations [3].

Regarding the brain biochemistry in schizophrenia, imbalances were identified mainly in dopaminergic neurotransmission [7], but also in serotonergic, cholinergic, and glutamatergic neurotransmission. Disorders in dopaminergic neurotransmission are represented by the hyperactivity of receptors in the mesolimbic tract (responsible for positive symptoms such as hallucinations, delusions, disorganized behavior, disordered thinking and speech, and abnormal movements), hypoactivity of receptors in the mesocortical tract (responsible for negative symptoms such as alogia, avolition, anhedonia, asociality, blunted affect, catatonia, and cognitive deficits), and dopaminergic hyperactivity in the nigrostriatal and tuberoinfundibular tract. Regarding serotonergic neurotransmission, it was observed that serotonergic hypoactivity causes dopaminergic hyperactivity, and, implicitly, the appearance of positive symptoms and serotonergic hyperactivity causes dopaminergic hypoactivity, resulting in negative symptoms [8]. Acetylcholine and glutamate transmission dysfunctions are responsible for the occurrence of cognitive impairment and some situations in which apoptosis occurs [3,9].

GPR genes are a large family of genes that encode G protein-coupled receptors (GPCRs), with over 800 members present in humans. Over half of the 800 receptors have sensory functions that mediate olfaction (400), taste (33), light perception (10), and pheromone signaling (5). More than 90% (370) of the remaining receptors are expressed in the brain. [10] The structure of GPCRs typically consists of seven transmembrane domains connected by extracellular and intracellular loops. The N-terminus of the protein is located on the extracellular side of the membrane, while the C-terminus is located on the intracellular side. The role of GPCRs is to transmit signals from various extracellular ligands, such as hormones, neurotransmitters, and sensory stimuli, to the inside of the cell. When a ligand binds to the GPCR on the extracellular side of the cell, it induces a conformational change in the receptor that activates an associated G protein on the intracellular side. This, in turn, triggers a signaling cascade that can lead to various cellular responses, such as the opening or closing of ion channels, activation of intracellular enzymes, and gene expression changes. GPR genes are expressed in a wide variety of tissues and organs throughout the body and play a critical role in many physiological processes, including sensory perception, neurotransmission, hormone signaling, and immune response. Dysregulation of GPR gene expression or function has been implicated in many human diseases, including cancer, cardiovascular disease, diabetes, and neurological disorders. Currently, almost 60% of currently marketed drugs target ~100 GPCRs directly or indirectly. The current catalog of all known GPCRs reveals that almost 100 GPCRs remain without known ligands and are labeled as “orphans”. The function of most members of this family remains elusive and is a largely unexplored reservoir of potential therapeutic targets [11].

The **Super Conserved Receptor Expressed in the Brain** (**SREB**) family is a group of related GPCRs, they have the characteristic of being extremely expressed in several regions of the brain, especially in the hippocampus [11]. Another interesting feature is their high degree of evolutionary conservation [12].

Because schizophrenia is a problem for both the patient’s individual functioning and integration into society and the medical system, the aim of this study is to identify and evaluate the correlations between genes encoding GPCRs and specific schizophrenia changes in the brain and their potential to be a prognostic factor for the response to psychotropic treatment or even therapeutical targets. Proper evaluation of these factors is essential to have a better therapeutic response with a lower rate of chronicity and thus improve the long-term prognosis.

## 2. Materials and Methods

### 2.1. Search Strategy and Study Selection

The research articles included in this study were obtained by three investigators through a survey of PubMed, Elsevier, and ScienceDirect databases (up to July 2022) with the following combination of keywords: “gpr schizophrenia”, “gpr psychiatry”, or “gpr psychosis”. The references from the selected articles were also analyzed to identify other relevant articles. 

### 2.2. Inclusion and Exclusion Criteria

Research articles’ inclusion criteria were as follows: (1) studies involving GPCR expression in brain regions associated with schizophrenia or psychosis, (2) studies evaluating the diagnostic or therapeutic potential of different GPCRs in psychiatric disorders, and (3) studies published in the English language. Research articles exclusion criteria were as follows: (1) non-original papers, such as conference abstracts and letters to editors, (2) duplicate studies, and (3) papers not written in the English language. This was a manual survey; the first search using the keywords found the following 209 articles on PubMed, Elsevier, and ScienceDirect. Afterwards, the titles and abstracts published in the last 5 years were read, the number remaining being 81, the rest were eliminated for being unrelated, duplicates, and unavailable full texts. Following this, we excluded the articles that did not correspond to our inclusion criteria for being irrelevant to our studies’ hypothesis. Finally, our research team concluded that the 12 articles included in the final study met the inclusion criteria and contained sufficient and relevant information, this information is illustrated in Figure 1.

### 2.3. Data Items 

Following the analysis of the search results, all abstracts were evaluated and relevant articles were selected for full text survey. The data were collected and summarized for each gene, including the main outcomes or conclusion, positive and negative association with schizophrenia, or brain changes.

## 3. Results

### 3.1. The Final Studies Included

From the currently available studies regarding the potential association between GPR expression and schizophrenia, we identified twelve studies out of which two were on GPR-52 expression, four on GPR-85 expression, four on GPR-88 expression, and one on GPR-139 expression. In evaluating these studies, we focused on the expression of GPRs in relation to neurochemical mechanisms, anatomical regions, and phenotypic aspects known to be involved in the pathophysiology of schizophrenia.

#### 3.1.1. GPR-52 

GPR-52 expression is located mainly in the brain, especially in the striatum. Other regions of the brain responsible for some manifestations of psychiatric illnesses and in which GPR-52 has been expressed are the medial prefrontal cortex (MPFC), habenular nuclei, and basolateral amygdaloid. Their distribution is associated with the distribution of dopamine receptors, being colocalized with D1 receptors in MPFC and with D2 receptors in the basal ganglia [13,14]. This pattern overlaps with the mechanisms present in schizophrenia in which D1 receptors in the cortex are hypoactivated, being associated with the appearance of negative symptoms, while D2 receptor hyperactivity present in the striatum is manifested by positive symptoms [15]. Most GPR expression in the prefrontal cortex (PFC) was glutamatergic and only 10% was GABAergic [16]. To verify the influence of this gene on behaviors, the reactions of knock-out (KO) and transgenic (TG) vs. wild-type (WT) mice to the administration of psychostimulants that mimic schizophrenia (NMDA receptor antagonists) were compared. The conclusion was that the KO group showed typical psychotic behaviors. Another hypothesis that suggests the protective role of GPR-52 is that this receptor is selectively and dose-dependently activated by reserpine, an antipsychotic known to relieve schizophrenic symptoms [13]. In another study, two GPR-52 agonists were found, one of which reversed rat-induced phencyclidine (PCP) hyperlocomotion in a dose-dependent manner and the second showed a robust, dose-dependent rescue of scPCP-induced deficits in the extra-dimensional shift phase of the attentional-set shifting task (ASST) and significantly rescued scPCP-induced deficits in social interaction at identical doses as in ASST without effects on object exploration or locomotor activity [14].

#### 3.1.2. GPR-85

GPR-85 expression was found mainly in the central nervous system, with the most intense expression being in the thalamus (paraventricular nuclei), hippocampal dentate gyrus, and indusium griseum. Its role in the development and differentiation of the central nervous system (CNS) is implied by the fact that it is present from the earliest stages of neurogenesis and is more intensely expressed in young neurons. In the adult central nervous system, low levels of transcripts were detected in the gray matter, but never in fiber tracts, pointing to a purely neuronal expression of GPR-85 [17]. Another study showed the direct influence of GPR-85 expression on brain size, behavior, and vulnerability to schizophrenia by comparing characteristics between transgenic mice and wild-type mice with knock-outs. The results showed that transgenic mice had a lower brain volume and an increase in the size of the cerebral ventricles, and the cortical neurons of these samples were smaller and more compacted, with reduced arborizations [18,19]; and 2 weeks after birth, they presented increased apoptotic levels. Social interaction, pre-pulse inhibition (PPI) [18], and contextual memory were affected in transgenic mice, while knock-outs had a discrete reduction in body weight compared to the transgenic type but an increase in the brain and cerebellum volume by about 10% and an improvement in contextual memory [18]; this was probably in the context of amplified neurogenesis in the hippocampus observed in this type of mouse [19]. In a family-based association test (FBAT) with human samples, two alleles of the GPR-85 gene were identified to be associated with schizophrenia, the presence of which is associated with a decrease in hippocampal cortex [18]. These risk-associated variants in SREB2 are associated with phenotypes similar to those found in patients with schizophrenia in the dorso-lateral prefrontal cortex (DLPFC) and the amygdala of males, while the pattern is opposite in females [20]. For transgenic mice, a significant decrease in desmoplakin, calbindin, and interleukin-1 genes was observed, which could disrupt the normal microenvironment necessary for neurogenesis in the dentate gyrus [19].

#### 3.1.3. GPR-88

The highest levels of GPR-88 transcripts were found in the striatum, olfactory tubercles, and nucleus accumbens [21,22], with this distribution pattern being similar to that of GPR-139 expression. Moderate levels were observed in the cerebral neocortex and at a lower intensity in the amygdala and hypothalamus [21]. GPR-88 protein expression in mice is more concentrated in the striatum than in the cortex and undetected in the cerebellum, and a similar distribution pattern was observed in monkeys (*M. Fascicularis*) [21]. In the striatum, GPR-88 is expressed in all neurons positive for substance P or enkephalin transcripts. The expression seems to be restricted to the somatodendritic compartment of medium spiny neurons (MSNs), which is similar to GPR-85 expression. This study highlights the existence of a functional relationship between dopamine receptors and GPR-88 expression, which is implied by the fact that L-dopa treatment led to the normalization of GPR-88 expression in striatopallidal and striatonigral MSNs in dopamine-depleted mice through D1 and D2 receptor-mediated mechanisms, respectively [21]. The role of GPR-88 expression in the pathophysiology of schizophrenia is also supported by the fact that its expression influences cognitive and motor functions, as demonstrated by the silencing of the GPR-88 gene in the nucleus accumbens, which reduces the typical manifestations of schizophrenia induced by amphetamine administration in the PCP-induced schizophrenia model in mice [23]. GPR-88 expression was directly associated with schizophrenia in a group study in South Africa [24]. In a study by Sheree F. Logue et al., it is suggested that GPR-88 expression modulates the function of the dopamine system, which is the main neurotransmitter involved in schizophrenia. They showed that although dopamine D2 receptor levels in GPR-88 KO mice are similar to those in WT animals, GPR-88 KO mice have lower basal extracellular levels of dopamine (DA) than WT but release the same amount of DA in response to amphetamine stimulation, suggesting a functional sensitivity to dopamine in GPR-85 KO mice. Obvious low sensitivity to haloperidol was observed in GPR-88 KO mice, with haloperidol being less effective in blocking climbing in GPR-88 KO mice than in WT mice; prepulse inhibition of startle response (PPI) is disrupted in GPR-88 KO mice (deficits in sensorimotor gating is a symptom of several psychiatric disorders and is exhibited by schizophrenic patients). D2 receptor antagonists reversed PPI deficiency in GPR-88 KO mice [22].

#### 3.1.4. GPR-139

GPR-139 expression has a distribution pattern in the CNS (olfactory tubercle, basal ganglia—including the caudate nucleus, putamen, and nucleus accumbens—thalamus, hypothalamus, substantia nigra, medulla, and pons) and the pituitary gland similar to the D2 dopamine receptors, and it is thought to be co-expressed with them in these areas, with the expression intensity being greater in favor of the dopamine receptors. This phenomenon has been found to be consistent between species, being observed both in rat and human tissue. It was proven experimentally that the two receptors present an interaction mechanism, with the GPR-139 gene being necessary for the activation of the D2 receptors by the dopamine agonist quinpirole in HEK293 cells, which could be reversed both through the use of dopamine antagonists and the GPR-139 antagonist [25]. In another study, enriched expression of GPR-139 in substance P-positive cells of the medial habenula in mice was found. To assess the role of GPR-139 expression in learning and cognition, KO animals were generated. As a result, KO mice appeared normal and performed comparably to wild-type animals in a range of standard tasks (learning, motivation, and social behavior); however, they were significantly impaired in models that reflect aspects of negative symptoms of schizophrenia, such as a progressive ratio, a measure of motivation, and nest-building, a model of self-neglect, and showed deficits in the novel object recognition model of working memory. A small molecule agonist was developed to test whether these deficits are reversible; it was tested in vivo in social interaction in poly (I:C) and cognitive tests in subchronic PCP models of schizophrenia, and it was observed to reverse these deficits, implying that GPR-139 expression plays a protective role in the development of negative symptoms in schizophrenia [26]. The final studies are summarized in Table 1; there we can find the main results along with the tissue distribution and their relationship with schizophrenia.

## 4. Discussion

In spite of the fact that the scientific literature involving the expression of GPRs is considerable in size and still growing, the current information regarding their involvement in schizophrenia is elusive and vague, and a considerable drawback could be the fact that most of the studies are carried out on mice, so interspecies differences and limits of the current schizophrenia models in mice should still be taken into account. Animal studies allow researchers to investigate the function of genes, proteins, and other biological components in a living organism; they provide essential insights into the biological mechanisms underlying various diseases and conditions. Transgenic mice mimic human diseases, enabling researchers to study disease progression, identify potential treatment targets, and evaluate therapeutic interventions. These studies contribute to the development of new treatments and the understanding of disease mechanisms.

Given the multitude of expressions of GPRs, it is to be expected that some of them could play a protective role and some could be part of the physio-pathological mechanism for a disease. We have found data about four expressions of GPRs involved in schizophrenia out of which two seemed to play a protective role, which was suggested by the better outcomes of the transgenic mice versus knock-out mice and also by the fact that agonists for these two GPRs reversed the chemically induced schizophrenia symptoms in mice. GPR-52 expression is colocalized with D1 receptors in the cortex and D2 receptors in the striatum, and it is proven to be mainly associated with glutamatergic cells in the PFC [13], which are known to have reductions in dendrite arborization, spine density, and synaptophysin expression across frontal and temporal regions in schizophrenia [27]. Other studies further imply that in schizophrenia, a decrease in NMDA receptor functioning with consequently elevated levels of glutamate is present [28]. Evidence is growing in the field of GPR agonists, with one study suggesting that there are already two distinct GPR-52 agonists that can reverse some of the schizophrenia-like symptoms induced by PCP administration [14]. Thus, GPR-52 expression may play a part in preventing negative symptoms through modulation of the D1 receptor function and could improve the NMDA receptor function, considering it seems to be a protective factor for schizophrenia and also associated with glutamate cells in brain areas known to be involved in this pathology. This could pave the way for new and more specific treatments, with higher efficiency in dealing with the negative symptoms that are so often overlooked even though they are the main factor involved in the disability of this pathology. GPR-139 expression has also been believed to play a protective role in schizophrenia, namely for negative symptoms, which is suggested by the fact that KO mice for this gene were significantly impaired in models that reflect motivation, self-neglect, and working memory, which are deficits reversed by the administration of a GPR-139 agonist [14]. The therapeutical effect of this agonist could be related to the dopamine D2 receptor, considering they are colocalized in many areas of interest to schizophrenia in the brain such as olfactory tubercles, basal ganglia, thalamus, hypothalamus, and substantia nigra, and it has been proven that they can interact in native tissue [25]. There are studies that conclude that GPR-139 is activated by L-Trp [29], which is known to be decreased in PCP-treated mice brain tissue [30] and also to have a decreased red blood cell uptake level in schizophrenic patients [31]. These findings could be a starting point for future studies about novel treatment methods involving L-Trp and its role in schizophrenia. However, the function of this receptor is intricate, and the role it plays in schizophrenia might be much more complex than that, considering that it is found in cells rich in substance P [26], which is believed to be involved in the pathophysiology of schizophrenia by modulating dopaminergic neurons in the mesocortical and mesolimbic pathways [32].

Out of the expressions of GPRs that were found to be associated with schizophrenia as risk factors, GPR-85 expression seems to play a role in delaying the development of the brain, as it has been found that Tg mice carrying the GPR-85 gene have lower brain volumes and cytoarchitecture disturbances in the cortex while the KO mice had increased brain volumes and neurogenesis in the hippocampus. The GPR-85 expression distribution showed an intense expression in the thalamus and the dentate gyrus of the hippocampus [12,18], with both regions being involved in the pathophysiology of schizophrenia [33,34]. Another GPR of interest to this pathology is the GPR-88 expression, which has been found mainly in the striatum in all neurons containing substance P but also in the olfactory tubercle and nucleus accumbens [35]. The olfactory tubercle has been proven to have smaller volumes in schizophrenic patients [22], while there are conflicting results about the volumes of nucleus accumbens in schizophrenia [36]. However, it has been proven that these neurons have a higher sensitivity to dopamine in schizophrenic patients [36,37,38,39,40,41]. For a more accurate image of the expression of GPCR in the brain, see Figure 2 below. More studies in this area are needed to elucidate whether this receptor is a cause of brain abnormalities in these regions or a consequence. The information found about GPR expression and mental disorders is scarce, and further studies are needed in this direction to better explain their role and potential as diagnostic or therapeutic instruments.

## 5. Conclusions

The current data about the expression of GPRs involved in schizophrenia is limited, and there is a significant heterogeneity regarding the samples and methods used, which makes comparing the results difficult. We have found that some of the expressions of GPRs are associated with an increased risk of developing schizophrenia; some were found to induce schizophrenia-like symptoms when translated to animals while others seemed to act like protective factors, providing greater resistance to transgenic animal subjects in chemically induced schizophrenia models. From these studies, we found there is promising evidence that suggests the relevance of GPRs as possible therapeutic targets, as more agonists for this type of receptor are being developed. Some of them are still considered orphans, and finding ligands for them could be of great importance in discovering their functions and how they interact with different receptors involved in mental disorders. Data found so far in specialized studies suggest that GPR-52 expression offers the most promising results. Significant results regarding GPR-52 expression show its distribution in key areas for schizophrenia development: the fact that KO mice show symptoms similar to psychosis but also the fact certain antipsychotics have a direct effect on them. Thus, they are a valuable target for more targeted studies and the development of new drugs aimed at schizophrenia symptoms, which can also target not just the positive symptoms but also the negative and cognitive ones. 

We believe that understanding the specific GPCR expression involved and their signaling pathways holds promise for the development of targeted therapies to treat this complex psychiatric condition.

## Figures and Tables

**Figure 1 pharmaceuticals-17-00085-f001:**
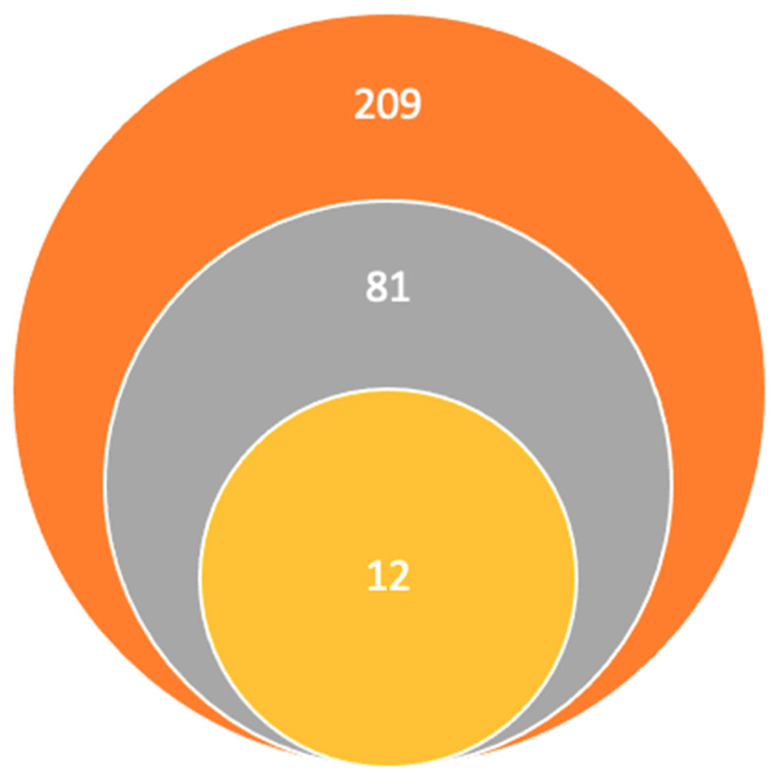
Study selection: 
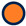
 Titles and abstracts published in the last 5 years using keywords. 
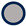
 Studies remaining after the elimination of other titles for being unrelated, duplicates, and unavailable full texts. 
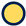
 Articles included in the final study.

**Figure 2 pharmaceuticals-17-00085-f002:**
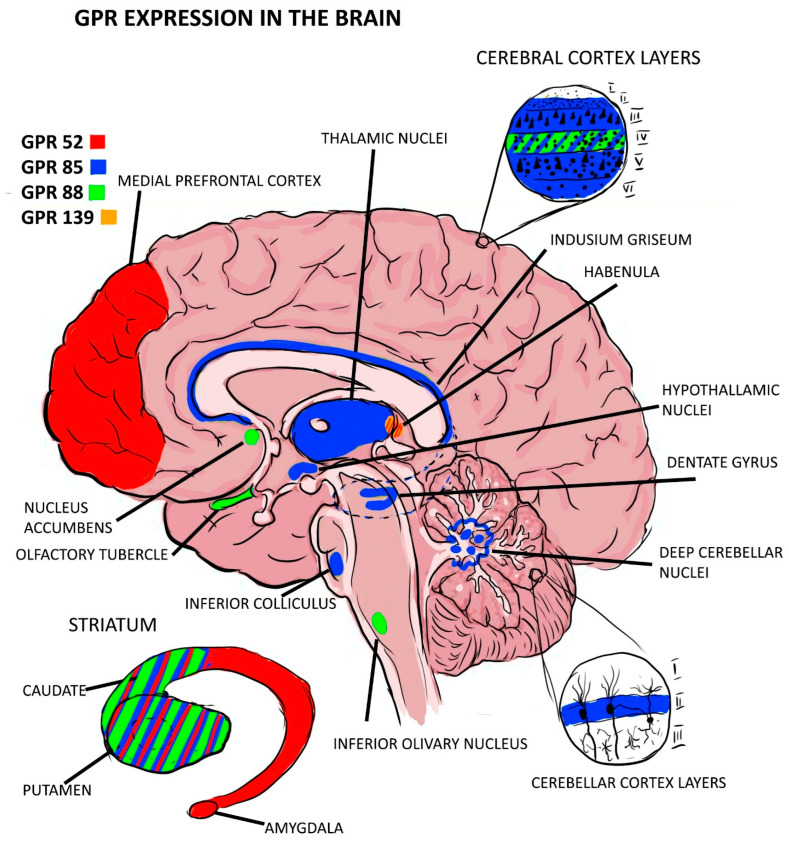
GPR expression in brain regions.

**Table 1 pharmaceuticals-17-00085-t001:** GPCR expression disturbances, dysfunction, or alteration associated with symptoms and signs of schizophrenia.

GPR	Main Results	Tissue Distribution	Relationship to Schizophrenia
GPR-52 [13,14]	Is a druggable Gs-coupled receptor; has a 22-residue ECL2 that folds into a small module and occupies the orthosteric binding pocket of the receptor, establishing interactions with transmembrane helices to maintain its configuration	Exclusively in the brain, especially in the striatum.	Potentially protective
	Agonists: 12c, 23a, 23d, 23e, 23f, and 23h Regulates various brain functions through activation of cAMPC-dependent pathways		
	KO group showed typical psychotic behaviors		
	GPR-52 receptor is selectively and dose-dependently activated by reserpine		
	GPR-52 agonists can reverse rat-induced phencyclidine (PCP) hyperlocomotion		
GPR-85 [17,18,19,20]	Is a protein-coding gene.	In the central nervous system, highest expression in the thalamus, while it was hardly detectable in spinal cord and the corpus callosum. Is more intensely expressed in young neurons.	Potentially risky
	Transgenic mice had a lower brain volume, an increase in the size of the cerebral ventricles, and the cortical neurons were smaller and compacted, with reduced arborizations.		
	After 2 weeks of birth, TG mice presented increased apoptotic levels		
	Human samples, two alleles of the GPR-85 gene were identified to be associated with schizophrenia and decrease in hippocampal cortex		
GPR-88 [21,22,23,24]	Is a protein-coding gene.	In the striatum, olfactory tubercles and in the nucleus accumbens.	Potentially risky
	Known agonists are: 2-PCCA, RTI-13951–33, and phenylglycinol derivatives.		
	Enables cytoskeletal motor activity. Involved in motor learning.		
	L-dopa treatment led to normalization of GPR-88 expression in striatopallidal and striatonigral MSNs in dopamine depleted mice through D1 and D2 receptor mediated mechanisms		
	Silencing of the GPR-88 gene in the nucleus accumbens reduces the typical manifestations of schizophrenia		
	Was directly associated with schizophrenia in a group study in South Africa Modulates the function of the dopamine system		
GPR-139 [26]	The aromatic amino acids could be the endogenous signaling molecules for GPR-139.	In the CNS and the pituitary gland similar to the D2 dopamine receptors.	Potentially protective
	Knock-out mice were significantly impaired in models that reflect aspects of negative symptoms		
	The small molecule agonist was observed to reverse deficits in models of schizophrenia including cognition in a subchronic PCP induced attentional set-shifting paradigm and social interaction		
	Plays a protective role in the development of negative symptoms in schizophrenia		

## Data Availability

Data sharing is not applicable.

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
