# Peer review of "Involvement of the Expression of G Protein-Coupled Receptors in Schizophrenia"

_pharmaceuticals, 2024, doi:10.3390/ph17010085_

Round 1

Reviewer 1 Report

Comments and Suggestions for Authors

I find the content of this review interesting. However, the overall structure of this review is very disorganized and the English is difficult to comprehend. When creating a table, first divide it into multiple columns. In the first column, briefly depict the structure of the receptor (7TM). If there are known agonists, also illustrate the structure of the agonist. Briefly describe the signal transduction pathway and include any receptors that are functionally related. In the following section, please provide a description of the tissue distribution. And in the other column, describe the relationship to schizophrenia. Typos: Potencially→Potentially, Symilar→ similar, Induseum grisium→ indusium griseum

Comments on the Quality of English Language

The content of this review is written in such a difficult manner that it is not easy to understand exactly what the author is trying to convey. I believe that native speakers should correct English.

Author Response

We thank the reviewers for their insightful observations. Herein, we present our point-by-point explanations for the revision of our manuscript.

The typos that were present have been corrected by a native speaker of English, and Table 1 has been restructured as per the suggestions offered.

We have simplified the information from the first column, we have given examples of agonists where they are known. Also, we have made an additional column where we describe the tissue distribution and a final column where we describe their relationship with schizophrenia.

We must also mention that we, as per the suggestion of Reviewer 2, we have renamed the table and it is now specified in the main body of the text. 

Thank you for your consideration.

Reviewer 2 Report

Comments and Suggestions for Authors

The topic is very actual, fitting in the scope of the journal. I appreciate, that is understandable and interesting also for clinically oriented psychiatrists.

I have only several small comments and suggestions.

  Ad table 1- I would appreciate a small correction:

1.      The name of the table  - to specify the content.

       2    In the first column there is GPR identification, so it is not necessary to repeat ties in   

            the second column, the text could be reduced  just to pick up important data on  

            separate lines ( location, data with KO and transgenic mice, results with human

           samples).

Ad table 2 – this is abundant, unclear I suggest deleting it- all data are given in the text.

This table could be available only on demand (as a support material).

Author Response

We thank the reviewer for the insightful observations, we present of our point-by-point explanations for the revision of our manuscript.

We have renamed Table 1 and specified its content in it, and we have mentioned the title in the main of the body of the text. Table 1 has also been simplified as per the suggestions from yourself and Reviewer 1. 

Also, as per the suggestions mentioned, Table 2 has now been removed from the review and is available as support material if needed.

Thank you for your consideration. 

Reviewer 3 Report

Comments and Suggestions for Authors

The present review approaches the possible involvement of some GPCRs in the pathophysiology and treatment oof schizophrenia. The article is clearly written and well-focused. However, there is some imprecision in the use of various terms and concepts that the authors should address to make the review sufficiently rigorous. Details follow here:

The authors state in various ways that GPCRs “are associated with” schizophrenia or that “GPCRs ….increase or decrease the risk for SCHZ”. However, those statements are inadequate for various reasons. GPCRs as such don’t do any of those actions, rather GPCR disturbances, dysfunction or alteration would be what may be associated with symptoms and signs schizophrenia. On the other hand, it is still entirely unclear what alterations of information processes are at the root of the many manifestations of schizophrenia. Accordingly, special care must be dedicated to use with the utmost precision words and concepts when attempting to explain possible mechanisms in which GPCR dysfunction might be involved.  

The paragraph starting with line 54 ha only one reference in the last sentence which is anoher review. Otherwise, it seems that many citations that should support some of the statements starting in at least lines 54, 56, 62 are missing.

Paragraph starting in line 94: Since arguably all humans carry the genes encoding for GPCRs what may be correlated with schizophrenia is not the genes, but the expression of those genes, some mutations they may carry or the turnover/processing of the corresponding proteins.

Figures 1 and 2 have a title but they dot have explanatory figure legends.

The figures are not mentioned anywhere in the main text of the manuscript either.

Section 3.1.1:

-“GPR 52 is located” should be changed to “GPR 52 expression is located”

-Several statements made in this section would require to be supported by appropriate citations.

Section 3.1.2:  “GPR 85 was found” should be changed to “GPR 85 expression was found”.

In general, is not correct to say that a gen is located or found somewhere in the brain, because what is found is its expression. Also, expression can refer to mRNA but also to the protein, and it is important to consistently specify what type of expression is being considered. If the authors are referring to the gen itself, they must specify so as well.

Comments on the Quality of English Language

English is adequate and understandable, though it could benefit from editing by a person fluent in neuroscientific English.

Author Response

We thank the reviewer for the insightful observations, we present our point-by-point explanation for the revision of our manuscript. 

We have corrected the terms and concepts used in the review as per the suggestions. Thus, we have mentioned GPR expression or GPR gene as needed in the entirety of the manuscript.

Also, references were added on lines 54-62 that support our statements on this paragraph. In paragraph 94 we have mentioned that GPR is involved with schizophrenia, not the gene.

We have added an explanatory legend for Figure 1 below the main figure, as suggested. Both figures are now properly referenced in the main body of the text. Regarding Figure 2, it presently has a legend explaining GPRs expression on each area of the brain and has not been modified from the original manuscript. Each color is explained in the top right corner of Figure 2. If further revisions of this table are necessary, kindly share your input with us and we shall modify it.

On section 3.1.1, we have changed the phrase “GPR 52 is located” to “GPR 52 expression is located”. Also, statements in this paragraph are now supported by appropriate citations.

For section 3.1.2 we have also changed “GPR 85 was found” to “GPR 85 expression was found” as per the previous suggestions.

Thank you very much for your consideration. 

Round 2

Reviewer 1 Report

Comments and Suggestions for Authors

Thank you for carefully revising the manuscript.